# Counterfactual generation for Out-of-Distribution data

Nawid Keshtmand*[1], Raul Santos-Rodriguez[1], and Jonathan Lawry[1]

[1]University of Bristol
{nawid.keshtmand,enrsr,J.Lawry}@bristol.ac.uk

## Abstract

Deploying machine learning models in safety-critical applications necessitates both reliable out-of-distribution (OOD) detection and interpretable model behavior. While substantial progress has been made in OOD detection and explainable AI (XAI), the question of why a model classifies a data point as OOD remains underexplored. Counterfactual explanations are a widely used XAI approach, yet they often fail in OOD contexts, as the generated examples may themselves be OOD. To address this limitation, we introduce the concept of OOD counterfactuals—perturbed inputs that transition between distinct OOD categories—to provide insight into the model's OOD classification decisions. We propose a novel method for generating OOD counterfactuals and evaluate it on synthetic, tabular, and image datasets. Empirical results demonstrate that our approach offers both quantitatively and qualitatively improved explanations compared to existing baselines.

## 1 Introduction

Machine learning (ML) models are increasingly deployed in predictive tasks, yet they often fail when encountering inputs from unfamiliar distributions. This vulnerability poses a major challenge for safety-critical applications such as healthcare and autonomous systems. Two key requirements for robust deployment are: (1) accurate detection of out-of-distribution (OOD) data—inputs drawn from distributions not seen during training—and (2) interpretability of the model's decisions, particularly regarding OOD classifications.

Although significant advances have been made in improving OOD detection performance [1], limited attention has been given to explaining why a model classifies a data point as OOD [2], despite growing ethical and regulatory demands for transparency [3]. OOD detection is typically framed as a binary classification problem, distinguishing in-distribution (ID) from OOD data. More nuanced formulations distinguish between *near-OOD* and *far-OOD* inputs [4, 5]. Far-OOD samples differ substantially from ID data, lacking both discriminative (class-specific) and non-discriminative (shared) features. In contrast, near-OOD samples share non-discriminative features with ID data but lack discriminative ones. For example, in image classification, backgrounds often represent non-discriminative features, while foreground objects are discriminative. A far-OOD image may differ in both, while a near-OOD image may share the background but contain unfamiliar objects.

This distinction is intuitive and can aid in explaining OOD classifications. Studying the transitions between in-distribution (ID), near-out-of-distribution (near-OOD), and far-out-of-distribution (far-OOD) regions offers valuable insight into a model's decision boundaries. Counterfactual explanations—describing how inputs can be minimally altered to change predictions—provide a natural framework for such analysis [6, 7]. However, existing counterfactual generation methods often break down in OOD scenarios, producing candidates that remain outside the training distribution unless explicitly constrained.

Our goal is to explain why a model/classifier trained on ID data identifies a given instance as OOD. To this end, we introduce a framework that distinguishes between discriminative and non-discriminative features to guide OOD samples toward more ID-like representations. By tracing how an OOD point can be perturbed to move from far-OOD to near-OOD and ultimately into the ID region, we both clarify the basis of the model's decision and generate counterfactuals that are more faithful to the ID distribution.

Importantly, our goal is not to provide a comprehensive solution to all aspects of counterfactual explanation for OOD data. We do not incorporate other feasibility constraints such as sparsity, though these can be added into the objective function if desired. Likewise, we are not claiming that this is the optimal approach for generating counterfactuals as other methods to generate features, partitions the data and perturbing the data can be used. The method can be further refined—for instance, by modifying the loss function to include additional terms that improve feasibility, interpretability or disentanglement. Rather, we introduce this framework referred to as 'OOD counterfactual' (OOD CF) as a novel direction for counterfactual generation by systematically adjusting the non-discriminative and

---

*Corresponding Author.

Proceedings of the 7th Northern Lights Deep Learning Conference (NLDL), PMLR 307, 2026.

discriminative features of an OOD data point to bridge the gap between OOD and ID spaces.

The main contributions of this work are:

1. A novel method, OOD CF, for generating counterfactual explanations for OOD data which explains why a model/classifier cosiders a data point as OOD.

2. Experiments demonstrating that our method produces counterfactuals that are more realistic and less likely to be identified as OOD compared to baseline approaches.

# 2 Background and Related Work

Throughout this paper, scalar mathematical quantities will be represented in lowercase, vectors will be in lowercase and bold, and matrix quantities will be capitalized and bold. The term 'class labels' will refer to the labels for the different classes of the ID dataset. The term 'OOD label' will refer to whether the data point is ID or OOD. The terms counterfactual and counterfactual explanation will be interchangeably used.

## 2.1 Counterfactual explanations

A common approach to interpreting machine learning (ML) model predictions is through counterfactual explanations, which articulate causal reasoning such as: "If X had not occurred, Y would not have occurred." This involves considering hypothetical scenarios that contradict the observed facts [6]. Given an input vector $\boldsymbol{x}$, we may ask why it was classified as class $c$, or why it was identified as OOD. Counterfactual explanations reframe this inquiry as: "What is the smallest change to $\boldsymbol{x}$ that would alter its classification?" Formally, a counterfactual explanation is an altered input vector $\boldsymbol{x'} = \boldsymbol{x} + \boldsymbol{\delta}$, where $\boldsymbol{\delta}$ represents a perturbation applied to $\boldsymbol{x}$. This perturbation typically moves the input across a decision boundary, resulting in a different classification [8]. The desiderata for counterfactual explanations vary across models and application domains. A fundamental property is realism—ensuring that the resulting counterfactual reflects a plausible scenario for the user [9]. Another widely valued property is sparsity, which promotes interpretability by limiting the number of features that change. When many features shift simultaneously, it becomes harder to attribute the classification change to specific variables. Block sparsity, where related features are modified together, can also enhance interpretability. Additional desirable attributes include feasibility (ensuring the counterfactual is attainable in practice), proximity (minimizing the magnitude of the perturbation), and data manifold closeness (ensuring that $\boldsymbol{x'}$ remains within the data distribution). These properties are often formulated as optimization objectives or constraints—for example, applying $L_1$ and $L_2$ regularization to $\boldsymbol{\delta}$, or constraining $\boldsymbol{x'}$ to remain close to a representative member of the target class.

One of the earliest and most influential approaches to counterfactual generation is the Counterfactual Instances (CFI) method proposed by Wachter et al. [8]. This method perturbs data points in the input space to change their predicted class, while encouraging sparse perturbations via $L_1$ regularization. Let $\boldsymbol{x}$ denote the original input instance and $\boldsymbol{x'}$ the corresponding counterfactual. Define $\hat{P}_t(\boldsymbol{x'})$ as the predicted posterior probability of target class $t$ given $\boldsymbol{x'}$, and $P_t$ as the desired probability (typically set to 1). The counterfactual is obtained by minimizing the following loss function:

$$\mathcal{L}(\boldsymbol{x'}, \boldsymbol{x}) = (\hat{P}_t(\boldsymbol{x'}) - P_t)^2 + \lambda L_1(\boldsymbol{x'}, \boldsymbol{x}) \quad (1)$$

where $L_1$ is defined as:

$$L_1(\boldsymbol{x'}, \boldsymbol{x}) = |\boldsymbol{x'} - \boldsymbol{x}|_1 \quad (2)$$

The first term incentivizes the counterfactual to be confidently classified as the target class, while the second term encourages sparsity in the perturbation by minimizing the $L_1$ norm of the change. This balance facilitates counterfactuals that are both effective and interpretable. Following this, there have been many extensions that can be used to enable the generation of counterfactuals with particular properties of interest for several different purposes. One property of interest is that the generated counterfactuals lie close to the data manifold. One approach that aims to generate counterfactuals close to the data manifold is the Contrastive Explanation Method (CEM) by Dhurandhar et al [10]. This approach includes an elastic net $L_1 + L_2$ regularizer which encourages the solution to be both sparse and close to the original instance. Additionally, Dhurandhar et al train an auto-encoder to reconstruct instances of the training set. They then include the reconstruction error of the perturbed instance as an additional loss term in the objective function. As a result, the perturbed instance lies close to the training data manifold.

One approach to ensure counterfactuals are close to the data manifold is to generate counterfactuals close to a representative member of a class, which was the aim of the Counterfactuals Guided by Prototypes (Proto) approach by Van Looveren and Klaise [11]. The Proto approach has a similar loss function to CEM but includes an additional term in the loss function that optimizes for the distance between the latent features of a counterfactual explanation

instance and the latent features of the 'prototypical' instance of the target class $\boldsymbol{proto_i}$.

# 3 Method

## 3.1 Overview

To explain the classification of a data point, it is useful to examine the decision boundary between classes. Similarly, OOD detection can be treated as a classification problem, where a data point $\boldsymbol{x}$ belongs to the ID, near-OOD, or far-OOD class. To understand why a data point is classified as OOD, we propose investigating the decision boundaries between far-OOD, near-OOD, and ID points using counterfactual explanations.

### 3.1.1 Counterfactual explanation generation

The interpretability of a counterfactual explanation depends on how clearly the changes to inputs can be understood. Perturbing multiple latent features simultaneously can lead to "information overload", causing confusion and reducing trust [12]. We argue that counterfactual explanations are more interpretable when changes are limited to individual latent features or focused on specific areas of interest, such as the background or foreground object. Therefore, we aim to restrict perturbations to one area of interest at a time, specifically targeting either non-discriminative or discriminative latent features in this work.

### 3.1.2 Counterfactual explanation generation for OOD data

We illustrate the distinction between traditional counterfactual explanations and our proposed method using a toy example in Fig. 1. Consider an ID dataset consisting of blue triangles and blue circles. A classifier with a horizontal decision boundary distinguishes between the data solely based on the shape. A pink hexagon represents a far-OOD data point, differing from ID data in both shape (discriminative feature) and color (non-discriminative feature).Traditional counterfactual methods, such as Watcher et al. [8], typically modify only the discriminative features to match a target ID class, assuming that non-discriminative features remain consistent across ID classes. As shown in Fig. 1(a), applying this approach to the pink hexagon changes its shape to an oval, resulting in a pink oval. While this crosses the decision boundary, the counterfactual remains OOD due to the mismatched color—a non-discriminative feature not present in the ID dataset.

In contrast, transforming an OOD data point into an ID one requires aligning both discriminative and non-discriminative features. Our method achieves this through a two-stage process. First (Fig. 1(b)), the pink hexagon becomes a blue hexagon by modifying its non-discriminative feature (color), transitioning from far-OOD to near-OOD. Second (Fig. 1(c)), the shape is adjusted to produce a blue oval—an ID point with both features aligned with the ID data. This staged approach ensures that the generated counterfactual is more likely to be considered ID, as it matches both feature types. It also improves interpretability by isolating changes to non-discriminative and discriminative features, offering clearer insights into how each influences the model's OOD classification.

**Definition** In the context of explaining why a data point is classified as OOD, we formulate counterfactual explanation generation in terms of a perturbation $\boldsymbol{\delta}$ which satisfies Eqn. 3:

$$D_{ID}(\phi_z(\boldsymbol{x} + \boldsymbol{\delta})) < D_{OOD}(\phi_z(\boldsymbol{x} + \boldsymbol{\delta})) \qquad (3)$$

where $D_{ID}$ and $D_{OOD}$ are metrics that represent how far the input is from being ID and OOD respectively, and $\phi_z$ is a latent feature extractor. The intuition is that adding the perturbation should lead to $\boldsymbol{x}$ being more ID and less OOD. However, as we do not have access to the OOD dataset during training time, we instead aim to find $\boldsymbol{\delta}$ which satisfies Eqn. 4:

$$D_{ID}(\phi_z(\boldsymbol{x} + \boldsymbol{\delta})) < D_{ID}(\phi_z(\boldsymbol{x})) \qquad (4)$$

We formalize an OOD counterfactual explanation as a perturbed data point that is generated in a two-stage procedure that involves changing the non-discriminative followed by the discriminative latent features. More formally, a latent vector $\boldsymbol{z}$ is separated into a non-discriminative part, $\boldsymbol{z_n}$, and discriminative part $\boldsymbol{z_d}$ i.e., $\boldsymbol{z} = (\boldsymbol{z_n}; \boldsymbol{z_d})$. Then we can satisfy Eqn. 4 by breaking it down into two stages given by Eqns. 5 and 6:

$$D_{ID}(\phi_z(\boldsymbol{x} + \boldsymbol{\delta_n})) < D_{ID}(\phi_z(\boldsymbol{x})) \qquad (5)$$

$$D_{ID}(\phi_z(\boldsymbol{x} + \boldsymbol{\delta_d})) < D_{ID}(\phi_z(\boldsymbol{x})) \qquad (6)$$

where $\boldsymbol{\delta_n}$ and $\boldsymbol{\delta_d}$ correspond to the perturbations to the non-discriminative and discriminative feature partitions, respectively. Eqn. 4 is satisfied by having $\boldsymbol{\delta} = \boldsymbol{\delta_n} + \boldsymbol{\delta_d}$. The relationship between the far-OOD, near-OOD and the ID class can be seen in Fig. 2.

## 3.2 General Process for generating Counterfactual explanations for OOD data

In order to generate the OOD counterfactual explanation, three steps are required: 1) Obtain latent features of the OOD data, $\boldsymbol{z}$. 2) Decompose $\boldsymbol{z}$ into

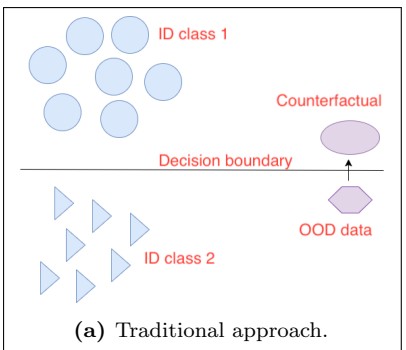
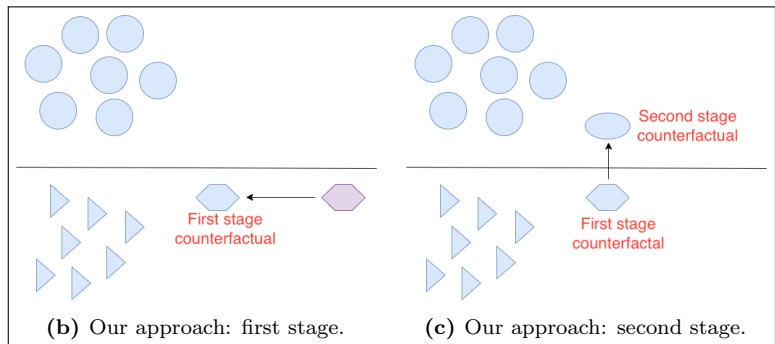

**(a)** Traditional approach.  **(b)** Our approach: first stage.  **(c)** Our approach: second stage.

**Figure 1.** A toy example with colored shapes illustrates how the OOD counterfactual approach differs from traditional methods. Traditional counterfactual approaches generate counterfactuals with shape (discriminative) features similar to the ID dataset. In contrast, our approach modifies the OOD data in two stages: first, by altering the color (non-discriminative) features, and second, by adjusting the shape (discriminative) features to ensure the final counterfactual matches both the shape and color features of the ID dataset.

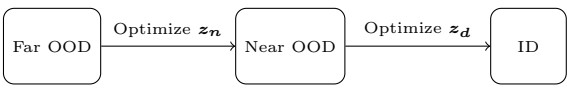

**Figure 2.** Relationship between Far OOD, Near OOD and ID data

.

two partitions, $z_d$ and $z_n$ using classifier of the ID dataset. 3) Perturb the OOD data, $x$, in a way that makes each latent partition more similar to those of the ID data using a density estimator.

**Obtain latent features of the OOD data, $z$.** To explain why a model considers a data point as OOD, we extract its latent features, $z$. Our approach is compatible with any model that produces latent features; such as a of a neural network classifier or dimensionality reduction techniques can be used such as UMAP (Uniform Manifold Approximation and Projection for Dimension Reduction) [13, 14]. However different feature extractors may be more or less suitable with different types of data and different modalities and so the feature extractor chosen may be context specific. For example, the latent features of an autoencodr may be effective for high-dimensional data where a non-linear feature space is required. In the case where a unsuperviseddimensionality reduction is used, a separate classifier will need to be used in the next step. In this work, we use Principal Component Analysis (PCA) to obtain $z$, as PCA captures features that maximize data variance.

**Decompose $z$ into two partitions, $z_d$ and $z_n$.** We explored several methods for partitioning the data and adopted a greedy approach to select a subset of features, $z_d$, that maximize classification accuracy. This approach assumes that discrimina-

tive features are those that improve classifier performance. Since greedy methods assume feature independence, it aligns well with PCA-derived features. Specifically, we used a Quadratic Discriminant Analysis (QDA) classifier and the sequential feature selector class from scikit-learn to identify the top-k features that maximize mean ID class label classification accuracy on the test data. These features form the discriminative partition, while the remaining $n - k$ features constitute the non-discriminative partition. In this case, we chose K to be half the total number of features where we make the assumption that there is an equal number of discriminative and non-discriminative features.

**Perturb the OOD data, $x$, in a way that makes each latent partition more similar to those of the ID data using a density estimator.** Counterfactual explanations are generated by perturbing OOD data so that the $z_n$ and $z_d$ values of the counterfactuals resemble those of ID data. Density estimation based OOD detection methods, such as the Mahalanobis Distance, perform well, suggesting that log-likelihoods effectively differentiate ID from OOD data. Building on this, we propose a loss function that minimizes the NLL of data points under a density estimator. Similar to the feature extractor aspect, our approach is agnostic of the density estimator used with several different options being available such as energy-based model and normalzing flows [15, 16]. Different design choices can be seen in Appendix A. In this case we chose a Gaussian Mixture Model (GMM) for its simplicity and good performance [17]. This involves modelling the feature space distribution using separate GMMs for $z_n$ and $z_d$ where the number of modes for $z_d$ is equal to the number of ID classes whilst we use a single mode for $z_n$. The generation process defined by Eqns. 5 and 6 involves two stages: first, counterfac-

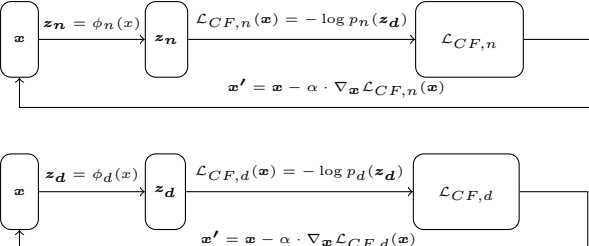

**Figure 3.** Optimization process to perturb the data to generate an OOD counterfactual explanation.

tuals are perturbed to minimize the NLL under the GMM defined by the non-discriminative partition. Then, they are further perturbed to minimize the NLL under the GMM defined by the discriminative partition. Although it is equally feasible to optimize for the discriminative partition followed by the non-discriminative partition. The loss for the non-discriminative, $\mathcal{L}_{CF,n}$ and discriminative partitions $\mathcal{L}_{CF,d}$ are given by Eqs. 13 and 14 respectively:

$$\mathcal{L}_{CF,n}(\boldsymbol{x}) = -\log p_n(\phi_n(\boldsymbol{x})) \qquad (7)$$

$$\mathcal{L}_{CF,d}(\boldsymbol{x}) = -\log p_d(\phi_d(\boldsymbol{x})) \qquad (8)$$

where $\phi_n(\boldsymbol{x}) = \boldsymbol{z_n}$, $\phi_d(\boldsymbol{x}) = \boldsymbol{z_d}$, $p_n$ and $p_d$ are the density under the non-discriminative and discriminative GMM respectively. We refer to the loss for the non-discriminative and discriminative partitions as the Non-discriminative (Non-dis) and Discriminative (Dis) loss respectively. By using the Non-dis and Dis losses, we can generate the OOD counterfactual $x'$ in two stages using Eqns. 9 and 10 respectively.

$$\boldsymbol{x'} = \boldsymbol{x} - \alpha \cdot \nabla_{\boldsymbol{x}} \mathcal{L}_{CF,n}(\boldsymbol{x}) \equiv \boldsymbol{x} + \boldsymbol{\delta_n} \qquad (9)$$

$$\boldsymbol{x'} = \boldsymbol{x} - \alpha \cdot \nabla_{\boldsymbol{x}} \mathcal{L}_{CF,d}(\boldsymbol{x}) \equiv \boldsymbol{x} + \boldsymbol{\delta_d} \qquad (10)$$

Where $\alpha$ is a weighing parameter which can be determined using cross-validation. The process of generating the counterfactual explanations can be seen in Algorithm C.1 in Appendix C. Additionally, a visual representation of the generation of an OOD counterfactual can be seen in Fig. 3. Also, we focus on these terms to see the effect of the partitioning of the features, additional loss terms could be added to enforce properties such as sparsity as seen in Appendix B.

## 4 2D Synthetic Experiment

**Task** Examining the generation of an OOD counterfactual in a 2D synthetic dataset where we can see the optimization trajectory of a single OOD data point during the two-stage process.

**Dataset** We consider a low dimensional (2D) dataset where we generate two Gaussian distributions with means (3,0) and (-3,0) with a tied covariance matrix $\boldsymbol{\Sigma} = 0.5\boldsymbol{I_2}$. There is also an additional OOD distribution which has a mean (0,2) and covariance $\boldsymbol{\Sigma} = 0.3\boldsymbol{I_2}$. PCA with two principal components is then used to extract the latent features that capture the most variance in the ID data. The setup of the synthetic dataset and the principal components can be seen in Fig. 4.

**Results** Using the principal components $\boldsymbol{pc1}$ and $\boldsymbol{pc2}$, an OOD counterfactual explanation is generated via Eqns. 13 and 14. The optimization trajectory of a single data point, shown in Fig. 4, starts at (0, 2). Initially, the point moves vertically along $\boldsymbol{pc2}$, which does not aid classification. In the second stage, it shifts horizontally along $\boldsymbol{pc1}$ toward the centroid of Class 1. This two-stage process highlights the significance of both non-discriminative and discriminative latent features in determining whether a data point is OOD.

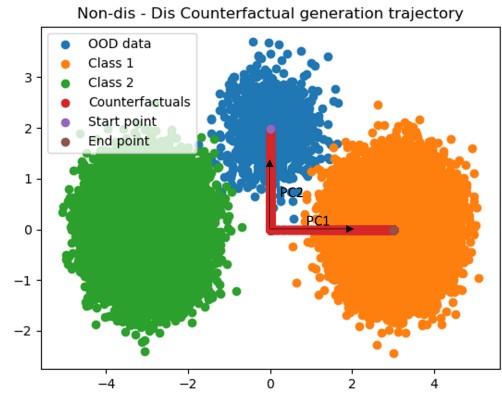

**Figure 4.** Optimization trajectory of a single OOD data point.

## 5 Tabular Experiment

**Task and Evaluation Metrics** We evaluate the effectiveness of our OOD CF approach by generating OOD counterfactual explanations for OOD images. The objective is for these counterfactual explanations to resemble data in the ID dataset. To assess performance, we measure the realism and minimality of the counterfactual explanations. Minimality is quantified using the $L_1$ distance between the original data points $\boldsymbol{x}$ and counterfactual explanations $\boldsymbol{x'}$.

Realism, a less precisely defined concept, is interpreted here as similarity to ID data in both discriminative and non-discriminative latent features. We quantify this by computing the AUROC between ID

data (positive examples) and counterfactual explanations from OOD data (negative examples). Detection scores for ID and OOD data points are derived using the Mahalanobis Distance, a simple yet effective scoring function, which is competitive with state of the art OOD detection approaches which is used frequently for detecting OOD data points for datasets like MNIST, SVHN and CIFAR100. The AUROC, which represents the Area Under the Receiver Operating Characteristic curve, reflects the probability that a positive example has a higher detection score than a negative example. In our case, lower AUROC values indicate counterfactual explanations that are less distinguishable from ID data and thus more realistic. Unlike traditional AUROC usage in OOD detection, lower values are desirable for this metric in our work. Additionally, we evaluate how well different approaches optimize the Non-Discriminative and Discriminative losses (Eqns. 13 and 14). Unless otherwise stated, metrics are averaged over four trials using unique random seeds, with a random subset of training data for each trial. For all reported metrics, lower values indicate better performance. In results tables, the best-performing method is bolded, and statistically significant improvements (p < 0.05, Wilcoxon signed-rank test [18]) are marked with an asterisk (*).

**Datasets**   We perform our analysis on three tabular datasets: Wine, Pima Diabetes, and Thyroid dataset. We use tabular datasets as this type of data is frequently used in the interpretability literature. To be able to use the OOD counterfactual explanation generation approach, the datasets needed to be preprocessed in some manner to have different classes and anomalous data points which can be treated as OOD for counterfactual explanation generation. Examples of this would be data points which have feature values which are distinct from the rest of the dataset, or belonging to a different class and therefore will not have similar discriminative features to the other classes of the ID dataset (mimiking a near-OOD data point). Explanations of the datasets and the preprocessing can be seen in Appendix D.

**Results and Discussion**   We compared the OOD Counterfactual Explanation (OOD CF) approach with three state-of-the-art baselines commonly used in counterfactual explanation literature: Counterfactual Instances (CFI) [8], Counterfactuals Guided by Prototypes (Proto) [11], and the Contrastive Explanation Method (CEM) [10]. The baselines were implemented using the Alibi package [19].

For all baselines, we trained a neural network classifier with two hidden layers matching the input dimensionality and output classes corresponding to the in-distribution (ID) classes. The classifiers were trained for 500 epochs using the SGD optimizer, a batch size of 128, and a learning rate of 0.01.

Table 1 shows that the OOD CF approach achieves lower Dis and Non-Dis losses compared to the baselines, indicating superior optimization of these metrics. Notably, despite not explicitly minimizing the $L_1$ distance, OOD CF outperforms the baselines in $L_1$ distance across most datasets, except for the Wine dataset where all methods perform similarly. In contrast, the baselines explicitly regularize to minimize $L_1$ distance but achieve inferior results, suggesting that it is easier to move the data from OOD to ID than moving a data point to cross the decision boundary of a particular ID class. Additionally, the AUROC values for OOD CF are consistently lower, indicating that its counterfactual explanations are more similar to ID data than those produced by the baselines. This highlights that lower Non-Dis and Dis losses correspond to more realistic and plausible counterfactual explanations with reduced $L_1$ distances. The realism of counterfactual explanations correlates with how well the method accounts for the density of ID data points. The CFI approach performs the worst in generating realistic counterfactual explanations, as it disregards data density and focuses solely on crossing decision boundaries. Conversely, the Proto approach, which generates counterfactual explanations near class prototypes, implicitly improves realism by aligning with higher-likelihood regions. Our OOD CF approach explicitly maximizes the likelihood of counterfactual explanations, leading to the most realistic and effective results among the evaluated methods.

## 6   Image Experiments

**Task and Evaluation Metrics**   We evaluate the effectiveness of our OOD CF approach by generating OOD counterfactual explanations for OOD images. This involves designating one dataset as the ID dataset and another as the OOD dataset. The goal is for OOD counterfactual explanations to resemble images in the ID dataset, assessed quantitatively using the metrics from Section 5. Additionally, we qualitatively analyze counterfactual explanations from various ID-OOD dataset pairs to gain deeper

| Dataset | Method | Non-dis | Dis | L1 | AUROC |
|---------|--------|---------|-----|-----|-------|
| Wine | OOD CF | **3.58**\* | **5.36**\* | 7.64 | **0.34**\* |
| | Proto | 8.57 | 7.09 | **7.01** | 0.99 |
| | CFI | 123.18 | 72.88 | 9.42 | 1.00 |
| | CEM | 24.34 | 14.54 | 7.80 | 1.00 |
| Diabetes | OOD CF | **4.28** | 3.66 | **2.67**\* | **0.33**\* |
| | Proto | 4.32 | **3.15** | 4.76 | 0.50 |
| | CFI | 291.01 | 150.88 | 6.26 | 0.65 |
| | CEM | 8.24 | 5.47 | 5.68 | 0.61 |
| Thyroid | OOD CF | **3.22**\* | **3.90**\* | **4.73**\* | **0.39**\* |
| | Proto | 3.47 | 4.14 | 9.01 | 0.82 |
| | CFI | 17.36 | 95.87 | 10.30 | 0.90 |
| | CEM | 11.73 | 55.15 | 9.66 | 0.95 |

**Table 1.**  Counterfactual explanation results for the Wine, Diabetes and Thyroid datasets

insights into the approach.

**Datasets** MNIST is a dataset of images of handwritten digits between 0 - 9 without any texture or color. Kuzushiji-MNIST (KMNIST) is an additional replacement for the MNIST dataset (28x28 grayscale, 70,000 images), provided in the original MNIST format [20]. ColoredMNISTRed is a variation of the MNIST dataset where the blue and green channels are set to zero to make the MNIST digits red [21].

## Results and Discussion

| Datasets | Method | Non-dis | Dis | L1 | AUROC |
|---|---|---|---|---|---|
| MNIST(ID)-KMNIST(OOD) | OOD CF | 18.51* | 25.31* | 43.35* | 0.26* |
| | Proto | 26.01 | 34.53 | 58.95 | 0.40 |
| | CFI | 30.22 | 37.44 | 58.80 | 0.41 |
| | CEM | 29.00 | 37.45 | 59.09 | 0.41 |
| MNIST(ID)-CMNIST(OOD) | OOD CF | 29.45* | 40.39* | 42.66* | 0.52* |
| | Proto | 32.09 | 44.92 | 52.26 | 0.65 |
| | CFI | 35.39 | 48.27 | 51.49 | 0.66 |
| | CEM | 37.34 | 50.77 | 51.72 | 0.65 |
| KMNIST(ID)-MNIST(OOD) | OOD CF | 23.66* | 28.81* | 33.83* | 0.22* |
| | Proto | 25.55 | 31.17 | 57.46 | 0.28 |
| | CFI | 28.15 | 34.50 | 57.11 | 0.36 |
| | CEM | 26.55 | 34.87 | 57.56 | 0.30 |

**Table 2.** Counterfactual results of different approaches for different ID-OOD pairs on various metrics .CMNIST is short for ColoredMNISTRed

**Quantitative analysis** Table 2 shows similar results to the tabular case, with OOD CF-generated counterfactual explanations being more realistic and less perturbed than the baselines. This is indicated by lower AUROC and $L_1$ values for OOD CF compared to the baselines. Additionally, lower Non-dis and Dis values further support the hypothesis that optimizing for these losses leads to more realistic counterfactual explanations. These findings demonstrate that the OOD CF approach is effective for both lower-dimensional tabular datasets and higher-dimensional image datasets.

**Qualitative analysis** Figures 5–7 illustrate OOD counterfactual explanations across three target classes. Each row contains six images: the initial OOD input and its nearest ID neighbor, the first-stage counterfactual and its neighbor, and the final (second-stage) counterfactual with its neighbor. Nearest neighbors are determined in PCA space. By examining which features are added or removed across stages, we gain insight into why the original input is considered OOD. In Fig. 5, using MNIST as ID and KMNIST as OOD, the first-stage transition (left to third image) lightens the background—consistent with MNIST's brighter appearance. In the second stage, class-specific features emerge, such as loops or segment removal, transforming the characters into shapes resembling a zero, six, or nine. This suggests that the KMNIST samples are OOD due to darker

central regions and a lack of distinctive digit features (e.g., class-specific loops) typically found in MNIST. A similar trend is observed in Fig. 6, where

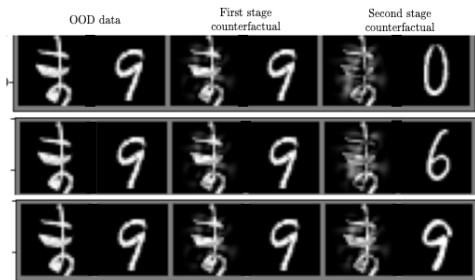

**Figure 5.** Each row contains six images: the initial OOD input and its nearest ID neighbor, first-stage counterfactual and its neighbor, and final (second-stage) counterfactual with its neighbor for 3 different classes for the MNIST(ID)-KMNIST(OOD) OOD CF.

ColoredMNIST (red digits) is the OOD set. In the first stage, grey patterns appear in the background, followed by class-specific structures in the second stage. The original ColoredMNIST points appear OOD due to the lack of white/grey central regions and missing discriminative features associated with specific MNIST digits. A limitation here is that the colored digit itself remains largely unchanged—our method focuses on adding plausible features rather than modifying existing ones. However, the effec-



**Figure 6.** Each row contains six images: the initial OOD input and its nearest ID neighbor, first-stage counterfactual and its neighbor, and final (second-stage) counterfactual with its neighbor for 3 different classes for the MNIST (ID)-ColoredMNIST (OOD) OOD CF.

tiveness of OOD counterfactual explanations can vary across ID-OOD dataset pairs. For instance, when KMNIST is used as the ID dataset, the first stage of counterfactual generation introduces irregular greying of the background, while the second stage further lightens it. This is likely due to the structural complexity of Japanese characters, which often extend toward the image edges—unlike MNIST digits, which are typically centered. This suggests that MNIST digits are identified as OOD partly because they lack sufficient brightness in the outer regions. However, class-discriminative changes are less evident in this setting. As shown in Fig. 7, second-stage counterfactuals (fifth column) optimized for different

target classes are visually similar, making it difficult to interpret class-specific transformations.

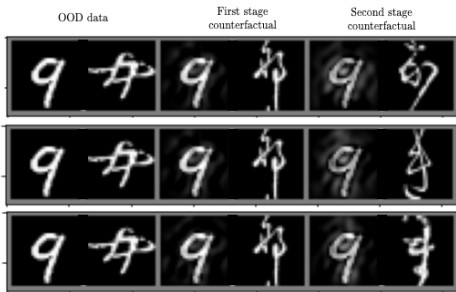

**Figure 7.** Each row contains six images: the initial OOD input and its nearest ID neighbor, first-stage counterfactual and its neighbor, and final (second-stage) counterfactual with its neighbor for 3 different classes for the KMNIST (ID)-MNIST (OOD) OOD CF.

In this case, it could be beneficial to use a different feature extractor such as neural network rather than PCA. Additionally, we compare our counterfactual explanations (Fig. 8) with those from the Proto approach, which performed best in the quantitative analysis (Fig. 9). The Proto baseline shows four images: the initial OOD point, its nearest neighbor in the ID data, the final counterfactual explanation, and the final counterfactual explanation's nearest neighbor. Although Proto removes part of the bottom loop to resemble a nine, it does not add the loop typical of a nine. In contrast, our approach successfully removes non-characteristic features (the top and bottom sections) and adds the loop, a key feature of a nine.

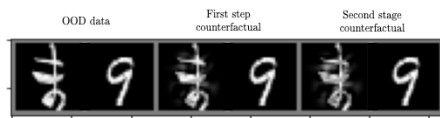

**Figure 8.** Each row contains six images: the initial OOD input and its nearest ID neighbor, first-stage counterfactual and its neighbor, and final (second-stage) counterfactual with its neighbor for a single class for the MNIST (ID)-KMNIST (OOD) OOD CF.

Overall, the counterfactual explanations generated by OOD CF better highlight the non-discriminative and discriminative features of the ID dataset more accurately than the baseline.

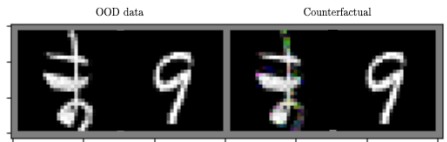

**Figure 9.** MNIST (ID)-KMNIST (OOD) counterfactual explanation generated by the Proto baseline.

# 7 Conclusion

This work addresses the challenge of explaining why data points are classified as OOD by introducing a novel framework for generating counterfactual explanations based on discriminative and non-discriminative latent features. We propose the concept of *OOD counterfactual explanations*, which transform OOD data points through a two-stage perturbation process: first aligning non-discriminative features, then discriminative ones, to produce more plausible ID data points. The method involves extracting latent features and partitioning them into non-discriminative ($z_n$) and discriminative ($z_d$) components, guiding structured transitions through far-OOD, near-OOD, and ID regions. Experimental results on both tabular and image datasets show that our approach outperforms baseline methods, yielding counterfactuals that are more realistic and better aligned with the ID distribution. This is reflected in reduced discriminative and non-discriminative losses, and lower AUROC scores when counterfactuals are evaluated using OOD detectors. Future work will explore improved techniques for feature partitioning, aiming to separate features into more interpretable categories. We hope our work will inspire further exploration of OOD explanations using non-discriminative and discriminative latent features.

# Acknowledgments

This work was supported by EPRSC PhD studentship as part of the Centre for Doctoral Training in Future Autonomous and Robotic Systems [grant number EP/L015293/1], EPSRC Impact Acceleration Account (EP/X525674/1), EPSRC LEAP Digital Health Hub (EP/X031349/1). RSR is funded by the UKRI Turing AI Fellowship [grant number EP/V024817/1].

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

# A    Alternative design choices

**Different ways to obtain non-discriminative and discriminative features** Alternatively to the method described in the main text, we can choose k (the number of discriminative features) by tracking how classification accuracy changes during sequential feature selection and selecting k at the point where accuracy begins to plateau. Another way to partition the data is to use the Independence cross-entropy loss described by Jacobsen et al. [22]. In their approach, k is set to the number of classes in the task, and the non-discriminative features are identified as those that maximize the cross-entropy loss, i.e., the features that are least predictive of the class label.

**Different density estimators** For Gaussian mixture models, each class corresponds to one Gaussian component, so the number of Gaussians equals the number of classes. This leads to a simple and generally stable optimization process, as gradients typically point toward the appropriate class-specific Gaussian. The likelihood also falls off away from each mean, reducing the chance of assigning high density to regions far from the training data.Energy-based model and normalizing flows are more complex and better fit the density of the data (potentially enabling the optimization to go follow a path of high density), however the optimization can be unstable due to the non-linear behavior of the density estimation and the models can incorrectly fit high density to areas far from the data.

# B    Additional loss terms

Additional terms in the loss function could be to add a L1 loss for the purpose of sparsity.where $L_1$ is given by Eqn. 11:

$$L_1(\boldsymbol{x}', \boldsymbol{x}) = \|\boldsymbol{x}' - \boldsymbol{x}\|_1 \tag{11}$$

$$L_{AE}(\boldsymbol{x}') = \|\boldsymbol{x}' - AE(\boldsymbol{x}')\|_1 \tag{12}$$

where an autoencoder reconstruction loss $L_{AE}$ can be an additional approach to capture the uncertainty in the data where AE denotes the autoencoder function. This can lead to an updated loss of:

$$\mathcal{L}_{CF,n}(\boldsymbol{x}', \boldsymbol{x}) = -\log p_n(\phi_n(\boldsymbol{x}')) + L_1(\boldsymbol{x}', \boldsymbol{x}) + L_{AE}(\boldsymbol{x}') \tag{13}$$

$$\mathcal{L}_{CF,d}(\boldsymbol{x}', \boldsymbol{x}) = -\log p_d(\phi_d(\boldsymbol{x}')) + L_1(\boldsymbol{x}', \boldsymbol{x}) + L_{AE}(\boldsymbol{x}') \tag{14}$$

---

**Algorithm C.1** Counterfactual generation - Natural language

---

**Require:** Test sample $\boldsymbol{x}$, target class $t$, training data $\boldsymbol{X}_{train}$
**Ensure:** Counterfactual explanation $\boldsymbol{x}'$
1: Extract latent features: $\boldsymbol{Z}_{train} \leftarrow$ FeatureExtractor($\boldsymbol{X}_{train}$)
2: Separate $\boldsymbol{Z}_{train}$ into $\boldsymbol{Z}_{n,train}$ (non-discriminative) and $\boldsymbol{Z}_{d,train}$ (discriminative)
3: Obtain discriminative features for class $t$: $\boldsymbol{Z}_{d,train}^t$
4: Fit a GMM to $\boldsymbol{Z}_{n,train}$ and $\boldsymbol{Z}_{d,train}^t$
5: $\boldsymbol{x}' \leftarrow \boldsymbol{x}$
6: $k \leftarrow 0$
7: **while** $k <$ max_iter **do**
8: $\quad \mathcal{L}_{CF,n} \leftarrow -\log p_n(\phi_n(\boldsymbol{x}))$
9: $\quad \boldsymbol{g} \leftarrow \nabla_{\boldsymbol{x}}(\mathcal{L}_{CF,n})$
10: $\quad \boldsymbol{x}' \leftarrow \boldsymbol{x}' - (\alpha \cdot \boldsymbol{g})$
11: $\quad k \leftarrow k + 1$
12: **end while**
13: $k \leftarrow 0$
14: **while** $k <$ max_iter **do**
15: $\quad \mathcal{L}_{CF,d} \leftarrow -\log p_d(\phi_d(\boldsymbol{x}))$
16: $\quad \boldsymbol{g} \leftarrow \nabla_{\boldsymbol{x}}(\mathcal{L}_{CF,d})$
17: $\quad \boldsymbol{x}' \leftarrow \boldsymbol{x}' - (\alpha \cdot \boldsymbol{g})$
18: $\quad k \leftarrow k + 1$
19: **end while**
20: **return** $\boldsymbol{x}'$

---

# C    Counterfactual Generation Algorithm

# D    Tabular datasets

**Wine**: These data points are the results of a chemical analysis of wines grown in the same region in Italy but derived from three different cultivars. The analysis determined the quantities of 13 constituents found in each of the three types of wines [23]. To define OOD data in this case, we make it so that all data points classified as class 2 are OOD.

**Diabetes**: This dataset consists of several medical predictor (independent) variables and one target (dependent) variable, Outcome. Independent variables include the number of pregnancies the patient has had, their BMI, insulin level, age, and so on. The outcomes are diabetic or non-diabetic [24]. To define OOD data in this case, we make it so that all data points with the age attribute above the upper quartile are OOD.

**Thyroid**: The problem is to determine whether a patient referred to the clinic is hypothyroid. Therefore three classes are built: normal (not hypothyroid), hyperfunction, and subnormal functioning [25]. Moreover, we defined the subnormal functioning class as the outlier class and the other two classes

are inliers, because subnormal functioning has the lowest number of data points present in the dataset.

# E    Histograms

To assess the effectiveness of the two-stage counterfactual explanation approach in altering non-discriminative and discriminative features, we analyzed the log probability of data using different GMMs. We used MNIST as the ID dataset and KMNIST as the OOD dataset. We hypothesized that a high log probability associated with a class-agnostic GMM generated from the ID data would indicate data points with feature values similar to the non-discriminative features of the ID data. Similarly, a high log probability from a class-conditional GMM would suggest data points with feature values resembling the discriminative features of the ID data.

We compared the log probabilities of the original OOD data point, the counterfactual explanation generated after optimizing for the first partition (first cf), and the two-stage counterfactual (cf), as shown in Fig. E.1. The results show that after optimization, both the first cf and cf have higher log-likelihoods for both class-agnostic and class-conditional GMMs compared to the initial OOD data point. Furthermore, optimizing both partitions results in a higher log-likelihood than optimizing just one.

From the class-agnostic GMM (left side of Fig. E.1), we see that the log-likelihood difference between the initial OOD data point and the first-stage counterfactual is large, while the difference between the first- and second-stage counterfactuals is small. This suggests that optimizing the non-discriminative partition leads to significant changes in non-discriminative features, while the second stage does not affect these features.

In the class-conditional log-likelihoods (right side of Fig. E.1), the first- and second-stage counterfactuals show higher log-likelihoods than the initial OOD data points. The log-likelihood difference between the initial OOD data points and the first-stage counterfactuals is small, whereas the difference between the first- and second-stage counterfactuals is large, indicating that the second stage effectively changes discriminative features while the first stage has minimal effect on them.

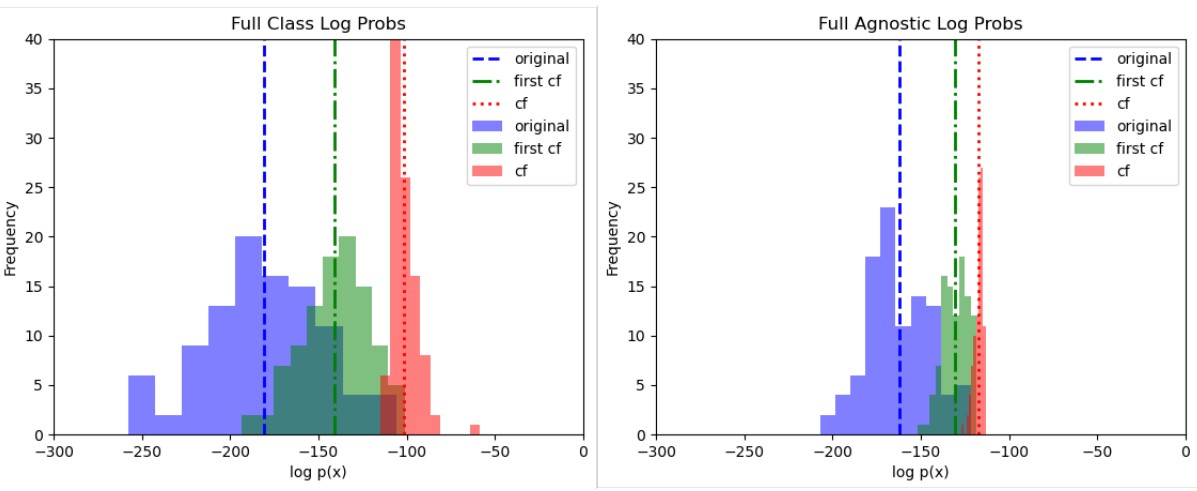

**Figure E.1.** Histograms of the log-likelihood for the counterfactual explanations generated for the MNIST-KMNIST ID-OOD dataset pair for a class-conditional (left) and a class-agnostic GMM (right).

