# OpenReview forum: "Counterfactual generation for Out-of-Distribution data"
_NLDL.org/2026/Conference — NLDL 2026 Spotlight_

### Official Review · Reviewer_hEGH · 2025-10-01
**Review for "Counterfactual generation for Out-of-Distribution data"**

**Rating:** 4
**Confidence:** 4

**Summary:**

This work proposes the framework ‘OOD counterfactual’ (OOD CF) to gain insight into why a model trained on in-distribution data (ID) might classify datapoints as out-of-distribution (OOD) using OOD counterfactual explanations.  This framework involves transforming the latent features of OOD data, $z$, in two stages. To begin with, $z$ is partitioned into non-discriminative ( $z_n$) and discriminative ($z_d$) features. Within OOD CF, in the first stage, the far-OOD data is transformed into near-OOD by perturbing $z_n$ and then in the second stage, the near-OOD data is transformed to ID data by perturbing $z_d$. In each stage, the latent features are perturbed to make them as close to the ID distribution as possible.

The paper contributes towards improved insight into how a model classifies data as OOD and the influence of non-discriminative and discriminative features towards the classification decision. Further, the work demonstrated both qualitatively and quantitatively how OOD CF can generate counterfactuals which are more realistic and faithful to ID data compared to existing baseline approaches.

**Strengths:**

I think the paper explored an interesting idea and was overall well-written. The motivation, explanation of core ideas and methodology were clear and technically sound. Results are also well-presented.

**Weaknesses:**

Some things to consider: I) The authors state they do not consider sparsity constraints as some prior works have done and it can be added to the objective function of this work if desired. Maybe it can be better motivated as to why sparsity was not included here.
II) In line 085 within the Introduction, something may be weird with the sentence "to generate features, partitions the data and perturbing the data can be used".
III) In line 045, the non-discriminative features are indicated as shared features while the discriminative features are class-specific. I found the mention of "shared" a bit confusing and vague in this context.
IV) The word "desiredata" in line 132 of Background and Related Work might be incorrectly written.
V) In equation 4, some more details, as were presented for equation 3 so as to explain the objective and terms, would be helpful to explain why this adapted formulation was used.
VI) The NLL abbreviation was likely not defined/written out in full before its first mention in line 357.
VIi) The start and end points in Figure 4 are a bit hard to distinguish; maybe different-shaped markers can help?
VIII) In the Datasets section, the KMNIST can be introduced by including that it has Japanese characters to add details to just mentioning "additional replacement for the MNIST dataset".
IX) As the authors mention, Figures such as Figures 6 and 7 with qualitative examples have subtle differences and are a bit hard to distinguish visually.

**Justification:**

This work presents an innovative and interesting step towards more realistic counterfactuals using OOD counterfactuals. The paper is sound and provides new insights into why models classify certain data as OOD and how the discriminative and non-discriminative features influence this. The two stage process proposed provides improved interpretability as well.

---

> ### Author Rebuttal · Authors · 2025-10-22
>
> Thank you for reading our work and providing valuable feedback. We are happy to see that you found this work interesting and well-written.
>
>
>
> > The authors state they do not consider sparsity constraints as some prior works have done and it can be added to the objective function of this work if desired. Maybe it can be better motivated as to why sparsity was not included here.
>
> We did not include sparsity in the loss function as we wanted to focus solely on the main idea of the two-stage procedure in the paper to see the effects of it without any additional loss terms. We will aim to make this clearer in the final version of the paper.
>
> > II) In line 085 within the Introduction, something may be weird with the sentence "to generate features, partitions the data and perturbing the data can be used". III) In line 045, the non-discriminative features are indicated as shared features while the discriminative features are class-specific. I found the mention of "shared" a bit confusing and vague in this context. IV) The word "desiredata" in line 132 of Background and Related Work might be incorrectly written.
> VI) The NLL abbreviation was likely not defined/written out in full before its first mention in line 357
> VIII) In the Datasets section, the KMNIST can be introduced by including that it has Japanese characters to add details to just mentioning "additional replacement for the MNIST dataset".
>
> Thank you for spotting these, we will make the suggested changes to improve clarity.

---

### Official Review · Reviewer_bGp9 · 2025-10-05
**This paper proposes a counterfactual-based framework to explain out-of-distribution detection decisions by perturbing latent features for improved interpretability.**

**Rating:** 4
**Confidence:** 1

**Summary:**

This paper proposes a counterfactual generation framework for out-of-distribution (OOD) detection that explains why inputs are classified as OOD by perturbing latent features in a structured way. The method aims to improve interpretability and robustness, showing promising quantitative and qualitative results on image and tabular datasets, though its novelty and experimental scope are limited.

**Strengths:**

This paper introduces a novel and well-motivated framework, termed OOD Counterfactuals (OOD CF), which generates counterfactual explanations for why data points are classified as out-of-distribution (OOD). The method’s key innovation lies in its two-stage perturbation process that sequentially adjusts non-discriminative and discriminative latent features, providing interpretable insight into model behavior. The approach is conceptually clear and experimentally validated on both tabular and image datasets, showing consistent quantitative improvements over established baselines such as CFI, CEM, and Proto. The inclusion of both quantitative (AUROC, L1 distance) and qualitative analyses, supported by visual examples, strengthens the empirical evidence and demonstrates the interpretability and realism of the generated counterfactuals.

**Weaknesses:**

While the idea of separating discriminative and non-discriminative latent features for counterfactual generation is interesting, the overall methodological novelty remains moderate, as the implementation largely builds upon existing components like PCA, GMMs, and standard gradient-based perturbations. The feature partitioning step is heuristic and may not generalize well to high-dimensional or non-linear feature spaces. Experiments are limited to relatively simple datasets (e.g., MNIST, KMNIST, small tabular sets) and lack evaluation on more complex models or real-world OOD scenarios. Furthermore, the approach depends heavily on chosen feature extractors and density estimators, with limited discussion on their stability or sensitivity. Overall, the paper is a promising conceptual contribution but would benefit from stronger theoretical grounding and broader experimental validation.

**Justification:**

This paper presents a well-motivated approach that enhances the interpretability of out-of-distribution detection through counterfactual generation. By perturbing discriminative and non-discriminative latent features, the method offers intuitive insights into model behavior and produces consistent empirical results with clear visual evidence. While its novelty is moderate and experiments are limited to simple datasets, the work provides a meaningful empirical contribution to explainable and robust machine learning.

---

> ### Author Rebuttal · Authors · 2025-10-22
>
> Thank you for reading our work and providing valuable feedback. We are happy to see that you found this work novel, well-motivated and conceptually clear.
>
> > The feature partitioning step is heuristic and may not generalize well to high-dimensional or non-linear feature spaces.
>
> To address the comment, we will expand our discussion on the feature partitioning approach to discuss feature partitioning approaches which can deal with these scenarios.
>
> > Furthermore, the approach depends heavily on chosen feature extractors and density estimators, with limited discussion on their stability or sensitivity.
>
> We plan to expand on the discussion on the different density estimators, comparing how different density estimators would behave.

---

### Official Review · Reviewer_HyHp · 2025-10-07

**Rating:** 4
**Confidence:** 4
**Final Rating:** 4
**Final Confidence:** 4

**Summary:**

The paper investigates counterfactual explanations for OOD data and proposes an intuitive two step approach to generate them. The first step of the proposed approach aims to convert Far-OOD data into Near-OOD data, while the second stage aims to convert the Near-OOD sample to ID. The proposed scheme works by extracting the latent representation of the OOD sample (here done via PCA), then decomposes the representation into discriminative and non-discriminative features (via QDA), and then finally perturbs the sample by minimizing the NLL (using a GMM). The overall procedure is quite generic and different methodological choices can be made for the different stages.

**Strengths:**

The paper is overall well written, making use of toy and synthetic examples to clearly outline the methodological approach.
The approach to decompose the problem into Far-OOD->Near-OOD->ID is intuitive and novel and the methodological choices are sensible.
Empirical results on simple datasets demonstrate the potential of the proposed approach.

**Weaknesses:**

While the paper proposes a relative generic framework, the investigation of different methodological approaches within this framework is lacking. In particular, this concerns the process of obtaining the latent features as well as the greedy feature decomposition. Especially given the sub-par performance on relatively simple datasets in Sec. 6, it would have been interesting to see if leveraging for instance neural network based approaches could alleviate these shortcomings. In particular since the baselines also leverage neural network based classifiers. Other concerns are related to the greedy decomposition and the strong assumption to assume an equal number of discriminative and non-discriminative features.

Minor: Missing reference around Line 355.

**Final Justification:**

While the method at this point is still limited to very simplistic datasets, I believe it is an interesting direction to explore and could motivate follow-up work aiming to address some of the concerns raised in the weaknesses and allowing it to scale to more complex datasets. Thus, I am still leaning towards accepting this work.

**Justification:**

The paper proposes a new interesting approach to counterfactual generation for OOD data by splitting the generation process into meaningful sub-parts. The approach is quite generic, opening up for investigation of the different components and future work. While the approach provides limited performance on more challenging image-based datasets, the overall idea is sound and results on synthetic experiments and tabular data provide an experimental foundation that can be built upon.

---

> ### Author Rebuttal · Authors · 2025-10-22
>
> Thank you for reading our work and providing valuable feedback. We are happy to see that you found this work well-written and novel. Additionally, thank you for spotting the missing reference -- that will be fixed for the final version of the paper.
>
> > This concerns the process of obtaining the latent features as well as the greedy feature decomposition
>
> To address the comment on the process of obtaining latent features as well as greedy feature decomposition, we will provide a more in-depth discussion on the process of generating latent features.
>
> > Concerns are related to the greedy decomposition and the strong assumption to assume an equal number of discriminative and non-discriminative features
>
> we will broaden the discussion on the greedy feature decomposition where we will discuss the assumption of using an equal number of discriminative and non-discriminative features and talk about other appraoaches and assumptions which could be made instead.

---

### Official Review · Reviewer_sVBK · 2025-10-07
**Review of new direction for counterfactual generation to explain out-of-distribution samples**

**Rating:** 5
**Confidence:** 3
**Final Rating:** 5
**Final Confidence:** 4

**Summary:**

They propose a novel method for generating out-of-distribution (OOD) counterfactuals which are inputs that have been perturbed to transition between distinct categories of OOD (and then to in-distribution). They use a two step perturbation process that first aligns non-discriminative features and then discriminative ones to end up with counterfactuals that better resemble in-distribution examples. Then, they evaluate its performance on various types of data compared to state-of-the-art baselines. They say the purpose is not about making the most robust procedure for generating these counterfactuals, but about showing the promise of this novel direction for explaining why samples are considered OOD and inspiring future work.

**Strengths:**

1. Explains and gives examples for important concepts like near-OOD and far-OOD
2. Explains the novelty of the paper, states the limitations, and states possible future work
3. Uses a good analogy to clearly explain how their procedure differs from previous methods
4. Procedure can be used for various modalities of data
5. Utilizes statistical significance to confirm better performance
6. Outperforms multiple state-of-the-art baseline procedures
7. The direction of this work seems beneficial for better AI explainability with regard to OOD

**Weaknesses:**

1. Does not include important considerations in the framework such as sparsity
2. Did not give a reason for why they are able to make the assumption at the end of the “Decompose z into two partitions, z_d and z_n” section. The assumption they made was that there are an equal number of discriminative and non-discriminative features.
3. Some of the figures have very small font

**Final Justification:**

After seeing how the authors addressed each of my concerns, it has helped solidify my decision to propose to accept the paper. The methods in this paper can be used for many data modalities and outperforms various state of the art baseline procedures. It also explicitly mentions limitations and possible future work. I still believe this paper has potential for a big impact since counterfactual explanations are powerful AI explainability tools and this paper seems to give a new direction to explore to improve upon them.

**Justification:**

It shows a novel direction that performs better than state-of-the-art procedures that seems promising. They say it is meant to be the beginning of a new direction in explaining OOD classification for models so it may not need to address all desired considerations in this paper as long as it states limitations and future work (which it does). Counterfactual explanations are a powerful tool for AI explainability and this paper seems to give a whole new direction to explore to improve upon them which has potential for a big impact.

---

> ### Author Rebuttal · Authors · 2025-10-22
>
> Thank you for reading our work and providing valuable feedback. We are happy to see that you found this work novel and that the direction of work is beneficial for AI explanability.
>
> > Does not include important considerations in the framework such as sparsity.
>
> We did not explicitly discuss sparsity in the framework as the purpose is not to propose the most robust procedure for generating these counterfactuals, but to show the promise of this novel direction for explaining why samples are considered OOD. However, in the paper, we will give examples of additional terms to be added to the loss function (such as the L1 penalty) to show how the loss function could change if we wanted to include additional sparsity constraints.
>
> > Did not give a reason for why they are able to make the assumption at the end of the “Decompose z into two partitions, $z_d$ and $z_n$” section. The assumption they made was that there are an equal number of discriminative and non-discriminative features.
>
> We used an equal number of discriminative and non-discriminative features in our case as it was a simple approach and likely a fair assumption. Other work assumes that the number of discriminative features is equal to the number of classes in the dataset.
>
> An alternative approach could be to see the number of features such that the classification accuracy no longer increases by a certain amount. We will expand on why we made the assumption in the paper.
> > Some of the figures have very small font
>
> Thank you for pointing this out. We will increase the size in the camera-ready version.

---

### Meta-Review · Area_Chair_pEWp · 2025-10-30

**Recommendation:** Accept (Poster)
**Confidence:** 4

**Metareview:**

The paper presents a novel and intuitive framework for generating counterfactuals to explain out-of-distribution samples through a two-stage perturbation process: first, aligning non-discriminative features (Far-OOD to Near-OOD), and then, discriminative ones (Near-OOD to ID).

While some reviewers initially noted limitations in the experimental scope and the heuristic nature of the feature-partitioning assumption, the authors’ detailed rebuttal and commitment to addressing these points satisfied all concerns. The reviewers agree that the paper’s originality, clarity, and potential impact outweigh its current limitations and recommend acceptance.

---

### Decision · Program_Chairs · 2025-11-05

**Decision:**

Accept (Spotlight)

**Comment:**

We recommend an oral and a poster presentation given the AC and reviewers recommendations.

A spotlight presentation refers to a poster selected for an oral highlight but not designated as a full oral presentation per the AC’s recommendation.